# Further Advance of *Gambierdiscus* Species in the Canary Islands, with the First Report of *Gambierdiscus belizeanus*

**DOI:** 10.3390/toxins12110692

**Published:** 2020-10-31

**Authors:** Àngels Tudó, Greta Gaiani, Maria Rey Varela, Takeshi Tsumuraya, Karl B. Andree, Margarita Fernández-Tejedor, Mònica Campàs, Jorge Diogène

**Affiliations:** 1Institut de Recerca i Tecnologies Agroalimentàries (IRTA), Ctra. Poble Nou Km 5.5, Sant Carles de la Ràpita, 43540 Tarragona, Spain; angie.ducaebre@gmail.com (À.T.); greta.gaiani@irta.cat (G.G.); maria.rey@irta.cat (M.R.V.); karl.andree@irta.cat (K.B.A.); margarita.fernandez@irta.cat (M.F.-T.); monica.campas@irta.cat (M.C.); 2Department of Biological Science, Graduate School of Science, Osaka Prefecture University, Osaka 599-8570, Japan; tsumu@b.s.osakafu-u.ac.jp

**Keywords:** ciguatera, ciguatoxins (CTXs), *Gambierdiscus*, neuroblastoma cell-based assay (CBA), immunoassay, immunosensor

## Abstract

Ciguatera Poisoning (CP) is a human food-borne poisoning that has been known since ancient times to be found mainly in tropical and subtropical areas, which occurs when fish or very rarely invertebrates contaminated with ciguatoxins (CTXs) are consumed. The genus of marine benthic dinoflagellates *Gambierdiscus* produces CTX precursors. The presence of *Gambierdiscus* species in a region is one indicator of CP risk. The Canary Islands (North Eastern Atlantic Ocean) is an area where CP cases have been reported since 2004. In the present study, samplings for *Gambierdiscus* cells were conducted in this area during 2016 and 2017. *Gambierdiscus* cells were isolated and identified as *G. australes*, *G. excentricus*, *G. caribaeus*, and *G. belizeanus* by molecular analysis. In this study, *G. belizeanus* is reported for the first time in the Canary Islands. *Gambierdiscus* isolates were cultured, and the CTX-like toxicity of forty-one strains was evaluated with the neuroblastoma cell-based assay (neuro-2a CBA). *G. excentricus* exhibited the highest CTX-like toxicity (9.5–2566.7 fg CTX1B equiv. cell^−1^) followed by *G. australes* (1.7–452.6.2 fg CTX1B equiv. cell^−1^). By contrast, the toxicity of *G. belizeanus* was low (5.6 fg CTX1B equiv. cell^−1^), and *G. caribaeus* did not exhibit CTX-like toxicity. In addition, for the *G. belizeanus* strain, the production of CTXs was evaluated with a colorimetric immunoassay and an electrochemical immunosensor resulting in *G. belizeanus* producing two types of CTX congeners (CTX1B and CTX3C series congeners) and can contribute to CP in the Canary Islands.

## 1. Introduction

*Gambierdiscus* [1] species are marine benthic dinoflagellates that produce secondary metabolites such as ciguatoxins (CTXs) and maitotoxins (MTXs). CTXs are lipid-soluble polyethers [2], which are introduced in food webs when filter feeders and herbivorous organisms eat free-swimming microalgal cells, macroalgae, or substrates that are colonized by benthic dinoflagellates [3]. Then, CTXs are transferred, transformed, and bioaccumulated through the food webs. Humans can get poisoned after the consumption of CTX-contaminated fish or very rarely some invertebrates (crustaceans, gastropods, echinoderms and bivalves) and suffer a disease known as Ciguatera Poisoning (CP) [4].

CTXs activate voltage-gated sodium channels (VGSCs) of cells, resulting in intracellular sodium increase and causing the repetitive firing of action potentials [5,6]. As a consequence, a few hours after the consumption of CTXs, gastrointestinal symptoms appear, typically followed by cardiac and neurological disorders. The neurological symptoms can last weeks, months, and even years [7]. The number of people who suffer from the disease is unknown, mainly due to the variability of symptoms, which leads to misdiagnoses and under-reporting. Annually, it is estimated that about 10,000–500,000 people suffer from the illness [8,9]. Even though CP is one of the most relevant poisonings worldwide, so far, there is no specific treatment [8]. CP was typical from tropical and subtropical regions, but during recent decades, CP cases have increased [10,11] and they have appeared in temperate zones through the importation of tropical ciguateric fish [12] or by the consumption of local ciguateric fish [13,14]. Climate change could change the geographical distribution of the dinoflagellates and the migration patterns of ciguateric fish and contribute to the geographical expansion of CP or increasing population densities of CTX-producing species in temperate areas [15,16]. In Europe, outside the boundaries of endemic areas in intertropical climates, new CP cases appeared in the North Eastern Atlantic Ocean after the consumption of fish from the Selvagens Islands (Portugal) and the Canary Islands (Spain) [17,18]. In the Canary Islands, CP is an illness of concern. In one decade (2008–2018), more than one hundred people have suffered from CP [19]. To prevent CP cases, the local authorities of this area have implemented the neuroblastoma cell-based assay (neuro-2a CBA) [20] to evaluate the possible presence of CTXs in the flesh of certain species of fish through the assessment of CTX-like toxicity [21].

It should be noted that only a few *Gambierdiscus* species have been confirmed to be CTXs producers [22,23], the toxin production is often very low, and not all the species produce the same quantities of toxins [22,24,25]. Therefore, the composition of species in the local areas could be an indicator of the level of risk to catch a ciguateric fish. One of the main factors to explain the latitudinal presence of *Gambierdiscus* species is the temperature [26], but other factors could be involved.

The Canary Islands are a transition zone between the oligotrophic waters associated with the Canary Current (CC), which is the subtropical gyre of the North Atlantic Ocean, and the eutrophic waters produced by the upwellings of deep cold waters with high nutrients along the African coast [27]. The east part of the Archipelago is semiarid; it is influenced by aeolian dust from the African continent and by the cold waters from the African upwelling system [28]. In contrast, the west is more humid, with more oceanic conditions and a minor influence of the African continent and the upwellings [28]. These conditions cause a longitudinal oceanographic east–west gradient of productivity (≈100 g of carbon m^−2^ yr^−1^) and the sea surface temperature (SST) (1–2 °C), which could explain the geographical distribution of the *Gambierdiscus* species.

Regarding the presence of *Gambierdiscus* species, in the Canary Islands, *Gambierdiscus* sp. was reported in 2004 [29]. Afterwards, in 2011, *G. excentricus* was described as a new species [30]. After that, the new species *G. silvae* [31] and *G. excentricus* were considered endemic from the Canary Islands. During the last decades, several samplings in the islands showed the high biodiversity and the wide geographical distribution of the *Gambierdiscus* genus [32,33]. At present, six species have been recorded in the Canary Islands, *G. australes* [31], *G. belizeanus* (in the current study), *G. caribaeus* [33], *G. carolinianus* [33], *G. excentricus*, and *G. silvae* [31], and none of them is limited to the Canary Islands.

The present study reports for the first time *G. belizeanus* in the Canary Islands. Previously, *G. belizeanus* was reported in Belize in the West Atlantic Sea [34], in Cuba [35], Cancun, St. Barthelemy, St. Marteen, and St. Thomas [36,37] in the Caribbean Sea. Additionally, it was detected in the Saudi Arabia in the Red Sea [38] and in Australia [39], Malaysia [40], and Kiribati Island [41] in the Pacific Ocean. Referring to *G. belizeanus*, it is considered a low toxin producer [24,38]. Among *Gambierdiscus* species from the Canary Islands, the evaluation of CTX-like toxicity has revealed that *G. excentricus* is one of the highest CTX-producing species within the genus *Gambierdiscus* and the most likely contributor to CP in the Atlantic Ocean [24,25,30].

Globally, it is not understood what triggers CP cases [4]. To fully understand the process of CP, to elucidate factors that may trigger CP, and to prevent the cases, it is necessary to identify and monitor the ciguateric fish but also to identify the CTX-producing species, their distribution, their physiology, and their toxicity.

The current study aimed to characterize the biodiversity and the geographical distribution of the *Gambierdiscus* genus in the seven most important islands of the Canary Islands Archipelago and to evaluate the potential CTX production of *Gambierdiscus* species to complement previous studies. For that purpose, samplings in the seven big islands of the Canary Archipelago were performed between October 2016 and October 2017. Isolates of *Gambierdiscus* cells were brought into culture. The molecular identification and morphological characterization of cultures contributed to the new report of *G. belizeanus* in the Archipelago. In addition, CTX-like toxicity was evaluated for forty-one strains of four species (*G. australes*, *G. belizeanus*, *G. caribaeus*, and *G. excentricus*) with the neuro-2a CBA, and the production of CTXs by the *G. belizeanus* strain was analyzed by a colorimetric immunoassay and an electrochemical immunosensor. New strategies within the microalgal field for future research of CP in the Canary Islands are further discussed.

## 2. Results and Discussion

### 2.1. Molecular Identification

Fifty-two strains including four species (*G. australes* (*n* = 32), *G. excentricus* (*n* = 18), *G. caribaeus* (*n* = 1) and *G. belizeanus* (*n* = 1)) were identified using sequences of the LSU D8-D10 rDNA region. The results of the BLAST analysis were well supported by the trees obtained using the Maximum Likelihood (ML) and the Bayesian Inference (BI) methods. Figure 1 shows the topology of the ML phylogenetic tree with bootstrap support values (bt) and the posterior probability (pp) of BI analysis displayed at branch nodes. Topography with the two phylogenetic trees was very similar. In both trees, the strains of this study are well defined within their respective clades for *G. australes*, *G. excentricus*, *G. caribaeus*, or *G. belizeanus* with bt/pp values of 96/1.00, 100/1.00, 98/0.92, and 96/1.00, respectively. Further, *G. pacificus* species were split into two clades. One sequence is grouped in the ML tree with *G. lewisii* with a high bootstrap value (>70). In contrast, in the BI tree, the clade of *G. lewisii* with *G. pacificus* appears, but it is less well supported (0.83 pp). The sequence of IRTA-SMM-17-421 was in the *G. belizeanus* cluster. This sequence exhibited 99% of similarity by BLAST with the isolate RS2-B6 of *G. belizeanus* (KY782638) from the Red Sea [38]. The genetic distance or pairwise distance (p-distance) between those two sequences was 0.011 substitutions per site. The strain IRTA-SMM-17-421 jointly with RS2-B6 has a deletion of 121 bp as described previously in Catania et al. [38], and this may indicate that it could be a ribosomal pseudogene. After excluding the deletion, the p-distance between IRTA-SMM-17-421 and *G. belizeanus* sequences ranged between 0.002 and 0.019 substitutions per site.

### 2.2. Morphological Characterization

The depth and width were measured in 50 cells for each species: *G. australes*, *G. excentricus*, *G. caribaeus* and *G. belizeanus* using the Calcofluor White stain method under light microscopy. Measurements for each species are shown in Table 1.

Cell morphology can vary depending on the culture conditions, growth phase, and different genotypes [32,42]. The morphological characterization showed that some cells of *G. australes* in the present work were smaller than the original description (76.0–93.0 µm of depth, 65.0–84.0 µm of width) in Chinain et al. [43]. However, overall, values are according to measurements presented in Bravo et al. [32], Litaker et al. [44], Rhodes et al. [23]. For *G. belizeanus*, the minimum measurements of depth in the present study are in accordance with the original description of 53–67 µm in Faust [34]; but, the minimum value for width (described in Faust as length), is lower than the first description (54–63 µm of width). Moreover, the maximum value for depth and width are higher than the description of Faust [34]. *G. caribaeus* cells from the current study were bigger than the original description in Litaker et al. [44] and the cells of Bravo et al. [32]. In reference to *G. excentricus*, all measurements are in accordance with the original description Fraga et al. [30], and the values were similar to Bravo et al. [32] and Hoppenrath et al. [45].

#### Morphological Characterization of *G. belizeanus*

Cells were anterior–posteriorly compressed. The plate formula of *G. belizeanus* was Po, 4′, 0a, 6″, 6c, ?s, 5‴, 0p, 2″″ based on Fraga et al. [30]. The cells were heavily aerolated (Figure 2A–C). Limits of the thecae are well defined by intercalary bands. These two latter characteristics are typical of *G. belizeanus* [38,40,44]. The 2′ plate is rectangular (Figure 2A), and the 2′’’’ plate is pentagonal (Figure 2B). Figure 2D shows the apical pore plate (Po).

### 2.3. Distribution of the Gambierdiscus Species in the Canary Islands

The observation of the samples under light microscopy showed that cells of *Gambierdiscus* spp. were found at 21 stations of the 53 stations sampled in the seven islands in 2016 and 2017. *Gambierdiscus* cells co-occurred with cells of the genera *Coolia*, *Ostreopsis*, *Prorocentrum*, *Amphidinium*, *Karenia* and *Trichodesmium*, among others. Details of the islands and the number of identifications for each station are shown in Figure 3. Overall, *G. australes* was the most abundant and it was present in all the islands. *G. excentricus* was the second most abundant. It was present in four islands Gran Canaria, Tenerife, La Gomera, and La Palma, excluding the eastern islands (Lanzarote and Fuerteventura) and the western island (El Hierro). *G. caribaeus* and *G. belizeanus* were identified in El Hierro.

Before 2004, the Canary Islands were considered outside of the geographical distribution of the *Gambierdiscus* genus, and the first reports of the genus were considered the result of a recent arrival in the islands [30]. The first reports of the *Gambierdiscus* genus in the Canary Islands were occasional, in one or two islands, and they were obtained from very few samples [30,31]. The finding of the new species *G. excentricus* and *G. silvae* in the Canary Islands, and the rapid increase in the number of species in consecutive samplings combined with the occurrence of CP cases in the islands [13] elicited the alarm to conduct urgent systematic and wide samplings in the area in order to understand the distribution and origin of the dinoflagellates and their potential contribution to CP cases. Rodríguez et al. [33] collected samples in numerous stations in La Graciosa (Chinijo Archipelago) in March 2015 and in the big islands excluding La Palma and La Gomera in September and October 2015. In the study of Rodríguez et al. [33], numerous isolates were identified, and *Gambierdiscus* cell abundances were determined in the sampling stations. The authors found a high richness of the *Gambierdiscus* genus in the Canary Islands, reporting five species: *G. excentricus*, *G. silvae*, *G. australes*, *G. caribaeus*, and *G. carolinianus*. This unexpected richness revealed a more likely ancient origin of the genus, which was contrary to what had been previously considered as a recent introduction [46]. In October 2017, Bravo and collaborators [32] performed systematic samplings in La Gomera and La Palma and complemented the geographical distribution of the *Gambierdiscus* species, among other issues. To date, samplings in the Canary Islands have been performed during very specific temporal scales, mainly in periods from October to September. In the current study, systematic samplings were performed in a higher number of stations, including new ones in the seven big islands. It is important to remark that a general bias in the identification process for the presence of species in a given area occurs since identification is a result of success growth under laboratory conditions. In Rodriguez et al. [33], this factor was partially avoided due to the identification of single cells in addition to the identification of cultured strains. The current study has contributed to a better understanding of the biodiversity of the genus in the Canary Islands, while providing a review and update of the geographical distribution of the species. A new contribution of the current work is the first report of *G. belizeanus* in the Canary Islands Archipelago (El Hierro) and *G. australes* in La Palma.

*Gambierdiscus* species have distinct lower and upper thermal limits and optimal temperatures for their growth [47]. Additionally, they are considered to have a high intraspecific variable response in growth depending on the temperature, salinity, and irradiance [47,48], and the response could be influenced by the geographical origin of the isolate [48]. Even so, biodiversity in the islands could follow a geographical pattern depending on the maximum and minimum temperatures on each island. For instance, as mentioned before, the SST of the western part of the Archipelago is higher. Therefore, a priori, the west could be more suitable for the warmer tropical species. This phenomenon has already been observed in fish [49]. Overall, the range of SST during 1972–2012 in the Archipelago was 15.9 °C in March–April and 25.5 °C in August–October [50]. The optimal temperatures for *Gambierdiscus* species are between 20 and 28 °C, and the maximal growth temperatures usually are >25 °C (Tester et al., 2020). Hence, it is expected that high abundances and more diversity of *Gambierdiscus* species would be found during August–October.

The distributions of *G. australes* and *G. excentricus* observed in the current study were similar to those reported by Rodriguez et al. [33] and Bravo et al. [32]. Table 2 compiles the records of *Gambierdiscus* species in the Canary Archipelago from the literature together with the results of the current study. *G. australes* is present in all seven big islands. These data are consistent since *G. australes* is the *Gambierdiscus* species with the widest optimal temperature range: between 19 and 28 °C [26,51]. In the current study, its presence is not dominant in all the islands, being slightly less abundant in La Gomera. This observation is in accordance with the previous results of Bravo et al. [32] reported for this island, where *G. excentricus* was the dominant taxa followed by *G. silvae*, *G. caribaeus*, and *G. australes*. The second most dominant species in the islands is *G. excentricus*, although it is not reported in El Hierro. Physiological data have not been reported for *G. excentricus*, even though its distribution in the Canary Islands is coherent considering that El Hierro has the most tropical conditions and that *G. excentricus* is more commonly present in temperate areas [26,45,52]. *G. silvae* was recorded in the central islands: Tenerife, La Gomera, and Gran Canaria. Overall, the longitudinal distribution of *G. silvae* is quite broad, including the Caribbean Sea and the Atlantic Sea [33,52]. However, compared to other species, one *G. silvae* strain, originating from Caribbean Sea, showed a narrow range of tolerance to temperature, and the maximal temperature for growth was low (24.8 °C) [48]. This result should be contrasted with maximal temperature of other *G. silvae* strains, but these data are presently lacking. *G. carolinianus* is present only in Tenerife. In Kibler et al. [47], the responses to environmental factors among strains were highly variable, but globally, *G. carolinianus* was well adapted to low temperatures (15 °C). Additionally, optimal temperatures for growth were also low (21.8–27.9 °C) [48]. Finally, *G. caribaeus* was reported in La Gomera and El Hierro, while *G. belizeanus* was present only in El Hierro. Experimentally, strains of *G. belizeanus* and *G. caribaeus* exhibited a wide range of temperature tolerance [48], but their range of temperature for maximal growth is considered high, since their temperature ranges are 26.1–29.1 °C [48] and 25–31 °C [48,53], respectively. Thus, the geographical distribution of both species in the Canary Islands is in accordance with the high-temperature adaptation in the laboratory. Tenerife and La Gomera have the highest richness of *Gambierdiscus* species. Tenerife is the biggest island and, consequently, more different habitats could be available. Additionally, this location in the middle of the Archipelago provides intermediate conditions. However, there are some inconsistences when trying to explain the biodiversity in the islands according to temperatures. For instance, experimentally, *G. belizeanus* has an optimal temperature range similar to *G. caribaeus* reported in El Hierro and in La Gomera. In contrast, *G. belizeanus* has not been reported in La Gomera. This may indicate that the presence of species is not yet well recorded or that more important factors for *Gambierdiscus* species are still unidentified. For instance, the exposure of the station to wave action was an important factor influencing the variability of macroalgae in the Canary Islands [54]. Additionally, it is important to highlight that there is a high intraspecific variation in the physiologic response under laboratory conditions. In other words, the strains from the Canary Islands may differ from other strains tested in previous studies.

The prevalence of the *Gambierdiscus* isolates in the Canary Islands since 2004 shows a good adaptation of the genus to the conditions found in the Archipelago. During recent decades, oceans have suffered a warming trend, and the SST in the Canary Islands is projected to increase 0.25 °C decade^−1^ [56]. The SST between 1985–2018 in the Canary Islands did not surpass 26 °C [50], which is far from the lethal temperatures for *Gambierdiscus* species (≈31 °C) [47,57]. Globally, the abundances of *Gambierdiscus* in the Canary Islands could be higher during the next years influenced by rising temperatures. Nonetheless, in the easternmost islands, the trend of the upwelling is not clear [56], and how upwelling could affect *Gambierdiscus* cell densities under the influence of the SST in the eastern islands is still to be studied.

### 2.4. Evaluation of CTX-Like Toxicity with the Neuro-2a CBA

The evaluation of the CTX-like toxicity was conducted for 41 extracts from cultures harvested in late exponential-early stationary phase of the four *Gambierdiscus* species: *G. australes*, *G. excentricus*, *G. caribaeus* and *G. belizeanus*, using the neuro-2a CBA. Obtaining CTX purified standards from microalgae is a very difficult task, since the CTX production in microalgae is often very low. For this reason, the toxicological evaluation often is carried out with CTX standards purified from fish. It has been found that CTX4A, CTX4B, and CTX3C are produced by the alga, and they are oxidized to the analogs CTX1B, 52-epi-54-deoxyCTX1B, 54-deoxyCTX1B, 2-hydroxyCTX3C, and 2,3-dihydroxyCTX3C [58]. In the present study, the reference molecule standard was CTX1B [59], which is a CTX typically found in large carnivorous fish in the Pacific Ocean [59,60] and which has never been found in microalgae.

As expected, in each experiment, the standard CTX1B displayed a non-significant reduction of viability with O/V^+^ treatment, whereas a typical sigmoid curve was exhibited in the O/V^−^ treatment. The average of the maximum exposed concentration of cells without toxicity in the O/V^−^ treatment for *G. australes*, *G. excentricus, G. belizeanus* and *G. caribaeus* ranged between 2–201, 0.2–50, 160 and 6800 cells mL^−1^, respectively. The one-way ANOVA showed that differences of CTX-like toxicities among the *G. excentricus* and *G. australes* were significant (*p*-value < 0.01). Moreover, the one-way ANOVA test for the CTX-like toxicity for the islands between species was significant; differences were for Fuerteventura for *G. australes* and in la Palma for *G. excentricus*. For these islands only one strain was tested. If these islands were not considered in the analysis, then the one-way ANOVA was not significant. Table 3 shows the results of the CTX-like toxicity expressed as fg CTX1B equiv. cell^−1^. The most toxic species was *G. excentricus* with a range of 9.5–2566.7 fg CTX1B equiv. cell^−1^. The most toxic strain was IRTA-SMM-17-330 from La Palma. The second most toxic species was *G. australes* with a range of 1.7–452.6 fg cell^−1^, followed by *G. belizeanus* which presented 5.6 ± 0.1 fg CTX1B equiv. cell^−1^. *G. caribaeus* did not show toxicity at 6800 cells mL^−1^ with an LOD of 0.42 fg CTX1B equiv. cell^−1^. Appendix A shows the neuro-2a cell viability and the CTX-like estimations obtained by the exposure to *Gambierdiscus* spp. extracts in O/V^+^ and O/V^−^ conditions.

The differences of CTX-like toxicity among strains of *G. excentricus* (*n* = 10) and *G. australes* (*n* = 29) from different islands are shown in a boxplot in Figure 4.

In parallel to the identification attempts of the *Gambierdiscus* species in the Canary Islands and the determination of their geographical distribution, efforts to confirm if the local populations of *Gambierdiscus* produce CTXs and contribute to the CP of the Canary Islands have been made. Currently, although the CTX-like toxicity has been described for almost all species from the Canary Islands [24,25,61], the confirmation of the CTXs production in isolates from the Canary Islands has only been determined for *G. excentricus* by Paz et al. [62]. The production of CTXs seems to be very low and infrequent compared to the production of MTXs [22]. For instance, maitotoxin-4 (MTX4) was detected in strains of *G. excentricus* from Tenerife (VGO791 and VGO792) by Pisapia et al. [63] and from la Gomera (IRTA-SMM-17-407) by Estevez et al. [64]. In the evaluation of CTX-like toxicity by neuro-2a CBA of the crude extracts, other compounds can interfere, since the evaluated toxicity in neuro-2a CBA is a composite effect, and sometimes, other compounds can cause unspecific mortality [25]. In Estevez et al. [64], the toxic effect on neuro-2a cells of the methanolic crude extract of the *G. excentricus* (IRTA-SMM-17-407) was very high, making the identification/quantification of CTX-like toxicity impossible. This effect was likely due to the presence of MTX4, which was also described in the same paper, since MTXs have high toxicity to neuro-2a cells [65]. In neuro-2a CBA, the use of controls with and without ouabain and veratridine allow the discrimination of CTX-like toxicity. Nevertheless, high amounts of MTXs, for example, and eventually other compounds, could also mask the presence of CTXs.

Fraga and collaborators [30] observed that *G. excentricus* from the Canary Islands produces compounds with high CTX-like toxicity. Afterwards, Pisapia et al. [24] reported that *G. silvae* was 100-fold less toxic and *G. australes* was 1000-fold less toxic than *G. excentricus*. Thus, *G. excentricus* and *G. silvae* seemed to be the top producers of CTX-like toxicity in the Atlantic Ocean. Nonetheless, after Reverté et al. [61] and Rossignoli et al. [25], this statement is controversial. Both studies listed *G. australes* and *G. excentricus* as the most toxic species in the Canary Islands and not *G. silvae*. In particular, Rossignoli et al. [25] examined the CTX-like toxicity using the neuro-2a CBA for the isolates of the five species originating from the Canary Islands. In that study, the highest CTX-like activity was for *G. excentricus* (VGO1361) with 510 fg CTX1B equiv. cell^−1^ followed by *G. australes* (VGO1252) with 107 fg CTX1B equiv. cell^−1^, *G. carolinianus* (VGO1197) with 101 fg CTX1B equiv. cell−1, *G. caribaeus* (VGO1367), with 90 fg CTX1B equiv. cell^−1^ and *G. silvae* (VGO1378) with 77 fg CTX1B equiv. cell^−1^. These results can be compared with our study because both studies used the same molecule of reference (CTX1B) and the same method for the toxicological evaluation.

Among the strains of the present work, *G. excentricus* exhibited the highest toxicity levels. These values are higher than those of Rossignoli et al. [25], being more similar to the levels measured by Fraga et al. [25] of 1100 fg CTX1B equiv. cell^−1^. The second most toxic species was *G. australes*. The mean values of the current study were similar to the highest values of Rossignoli et al. [25] (31-107 fg CTX1B equiv. cell^−1^) and globally lower than the values of Reverté et al. [61] (200 to 600 fg CTX1B equiv. cell^−1^). The *G. caribaeus* strain (IRTA-SMM-17-03) did not exhibit CTX-like toxicity; this in accordance with Rossignoli et al. [25]. In fact, in previous studies, *G. caribaeus*, *G. belizeanus*, and *G. carolinianus* were classified as low CTX producers [25,37]. *G. caribaeus* showed no CTX-like toxicity by CBA [66]. Referring to *G. belizeanus*, the CTX-like toxicity values of the current study are higher than the average of the toxicity for the *G. belizeanus* strains from the Red Sea of 0.038 fg of CTX1B equiv. cell^−1^ [38]. In the Red Sea, there is no confirmation of any CP cases [67], and *G. belizeanus* is the only *Gambierdiscus* species reported in that area. Even though only one isolate has been evaluated, its low toxicity, together with its restricted geographical distribution to El Hierro, suggests that the contribution of *G. belizeanus* to CP of the Canary Islands may be negligible, although more strains should be evaluated.

The literature shows that there is a high variation of CTX-like response between isolates within the same species [24,25,61]. This variation has been observed also in the current study. For *G. excentricus* and *G. australes*, the strains with the highest toxicity were 160 and 100-fold more toxic than the least toxic strain of the same species, respectively. This variability is higher than the obtained intraspecific values in Litaker et al. [37], Pisapia et al. [24], and Rossignoli et al. [25]. These differences may be related to the large number of strains that have been evaluated in the present study.

According to our results, the CTX-like toxicity levels have no clear pattern relating to the islands of origin. This is relevant because data of the fish from the official control program of ciguatera from the Canary Islands showed that ciguateric fish follow an east–west gradient. Toxic fish were more likely to be caught in Lanzarote (53%), followed by Fuerteventura (21%), Gran Canaria (18%), El Hierro (15%), Tenerife (14%), La Palma (5%), and La Gomera (2%) [21]. Sanchez-Henao et al. [21] suggested that the percentage of Lanzarote results must be considered with caution, since some samples were not accompanied by the total information and may not reflect the reality. Thus, the global tendency of ciguateric fish could be explained by the *Gambierdiscus* cell abundances from the samplings of Rodriguez et al. [33], which showed higher abundances in the east than in the west. However, for some fish species, the east–west gradient of CTX-like toxicity is not followed [21]. For instance, the major percentage of CTX positive groupers was found in El Hierro followed by Lanzarote and the other islands. As it has been mentioned before, the cell densities were lower in El Hierro than in Lanzarote and Fuerteventura. In addition, the presence of the most toxic species, *G. excentricus*, is not confirmed in El Hierro. Therefore, these results of ciguateric fish should be compared with cell abundances by seasonality and include temporal series for different years. Additional other factors could contribute to toxin production and bioaccumulation in fish, as well as the mobility of fish. After all, it is still early to establish a list of the riskiest areas for CP, as the relation between microalgae and fish are unknown in the Canary Islands, and further research should be undertaken.

### 2.5. Evaluation of the Presence of Two Series of CTX Congener Equivalents (CTX1B and CTX3C) in G. belizeanus with a Colorimetric Immunoassay and an Electrochemical Immunosensor

The role of the antibodies in the immunosensing tool is not to confirm the presence of individual CTXs congeners as in instrumental analysis techniques, but to screen the presence of compounds with wings structurally similar to those of the four CTX targets (CTX1B, 54-deoxyCTX1B, CTX3C, and 51-hydroxyCTX3C). Therefore, and because of the sandwich format of the assay, the analysis is indicating the presence of compounds structurally related to two series of CTXs congeners, although no evidence can be obtained about which specific CTXs congeners are present. It is important to mention that the strategy will detect only the CTXs recognized by the antibodies (the four major CTX congeners previously mentioned), but it is also important to highlight that the antibodies have been demonstrated to not cross-react with brevetoxin A, brevetoxin B, okadaic acid, and maitotoxins.

Analyses with the immunoassay and the immunosensor revealed the presence of the two series of CTXs congeners in the *G. belizeanus* extract. Immunoassay results when antibodies were used separately showed a higher concentration of 51-hydroxyCTX3C equiv. (0.28 ± 0.02 fg cell^−1^) than of CTX1B equiv. (0.15 ± 0.03 fg cell^−1^). As expected, the use of both antibodies simultaneously resulted in higher toxin content (0.40 ± 0.02 fg of CTX1B equiv. cell^−1^). Similar results were obtained with the electrochemical immunosensor. The use of 10C9 antibody resulted in a higher concentration (0.17 ± 0.08 fg 51-hydroxyCTX3C equiv. cell^−1^) than the one obtained with 3G8 (0.13 ± 0.03 fg of CTX1B equiv. cell^−1^). Again, the use of both antibodies together provided higher toxin content (0.35 ± 0.04 fg CTX1B equiv. cell^−1^). Both immunochemical tools provide similar CTX quantifications of CTX congeners in *G. belizeanus* extract. The use of these techniques not only revealed the presence of CTXs, but also allowed the discrimination between two series of CTX congeners (CTX1B and CTX3C). When using the mAbs separately, results showed slightly higher contents for the CTX3C series than for the CTX1B series. In fact, 51-hydroxyCTX3C equiv. contents were similar to those obtained in a previous study for *G. caribaeus* IRTA-SMM-17-03 (0.13–0.21 fg cell^−1^), *G. australes* IRTA-SMM-17-286 (0.16–0.37 fg cell^−1^), and *G. excentricus* VGO791 (0.16–0.31 fg cell^−1^), both originating from the Canary Islands [68]. Regarding CTX1B equiv. contents, the toxin contents in *G. belizeanus* were similar to those obtained for *G. caribaeus* IRTA-SMM-17-03 (0.13–0.24 fg cell^−1^) and surprisingly, *G. excentricus* strains IRTA-SMM-17-01, IRTA-SMM-17-407 and IRTA-SMM-17-432 (0.09–0.19 fg cell^−1^). It is important to note that CTX contents obtained with the immunochemical tools do not fully agree with those obtained with CBA. The reason is the different principle of recognition of both systems: whereas CBA is based on the toxic effect of a compound on the cell viability, the immunochemical tools are based on a structural recognition and affinity interaction between antibodies and their target molecules. Nevertheless, the analysis of microalgal extracts with different techniques provides complementary information and contributes toward improved characterization of the different *Gambierdiscus* species.

### 2.6. Future Research Strategies to Understand CP in the Canary Islands

During the last years, the official control of CTX-like toxicity evaluation of the harvested fish in the Canary Islands has been used as the first step to prevent CP cases [21,69]. The official control obliges all analyzed fish to be stored and frozen until CTX-like toxicity results are available, influencing its commercial value.

In order to prevent and to identify the future trends of CP in the Canary Islands, knowledge about the microalgal communities should be considered. The link between dinoflagellates and CP remains uncertain, and the key vectors of the transfer of CTXs into fish and eventually shellfish are still unclear. It is essential to identify the principal involved species. One of the first big steps in the microalgal field is the unequivocal identification of the species and the estimation of their abundance in the environmental samples. It is not clear whether only the genus *Gambierdiscus* can contribute to CP [22]. For example, *Coolia tropicalis* produces 44-methylgambierone (previously reported as MTX3) [70], which in human neuroblastoma cells (SH-SY5Y cell line) induces current sodium as the CTX3C but with lower potency [71]; additionally, neither of these toxins induce cell death in human cortical neurons when they were exposed at 20 nM concentration [72] Even so, a recent study considers it unlikely that 44-methylgambierone contributes to CP due to its low toxicity by mouse bioassay (MBA) [73]. It is necessary to clarify the toxin profiles of microalgae by instrumental analysis and analyze the toxicity of each compound to evaluate which species are low or high producers of CTX analogues. At present, there is a lack of knowledge of which compounds microalgae can produce and how these compounds interact with the food webs. Therefore, we need to identify these compounds in the microalgal, fish, and invertebrate matrixes and establish their interactions. Hence big gaps in the detection of fish and microalgal toxins still exist [74,75].

Another issue to be solved is related to the identification of *Gambierdiscus* species, which should be conducted to clarify the taxonomy and improve the molecular diagnoses. During the last years, there have been several attempts to identify *Gambierdiscus* species in environmental samples, and they have achieved good results [52,76,77], but they are not implemented in monitoring programs extensively.

The sampling method still requires standardization. Given that *Gambierdiscus* cells are found in a heterogeneous distribution, it is necessary to perform exhaustive geographical and temporal samplings using a standardized method. Artificial substrates have been used to sample benthic dinoflagellates such as *Gambierdiscus* spp., showing a reduction in the variability of densities of several samples collected at the same station [78]. However, when using artificial substrates, some ecological data are dismissed. For example, samplings in macroalgae substrates could aid in recognizing the potential preference by *Gambierdiscus* for particular macroalgae species or species assemblages. A priori, macroalgae are more visible and easier to monitor. Hence, understanding the population dynamics of the preferred macroalgae for *Gambierdiscus* could be relevant to explain the trend of the dinoflagellates. Additionally, identifying the grazers of these macroalgae could help to understand the accumulation through the food web. Nevertheless, it has to be taken into account that free-living cells could contribute to CP [3].

Additionally, to understand the future trend of populations in local areas, it is essential to identify critical factors for such regional populations. To this end, experimental design in field, such as environmental, ecological, and anthropogenic activities that potentially can modulate the *Gambierdiscus* populations, should be considered. This field data should be combined with experimental/laboratory data. Other factors should be examined to understand, for instance, which variables affect the toxin content of cells.

## 3. Conclusions

The present study provides more data of *Gambierdiscus* species distribution and toxicity in the Canary Islands. The new report of *G. australes* in La Palma and the new finding of *G. belizeanus* in the Canary Islands (El Hierro) show that data in the geographical and temporal scale is still scarce. Similar to the previous studies, the evaluation of the CTX-like toxicity shows that *G. excentricus* and *G. australes* are the species which could more contributeto CP. These species are widely distributed in the Canary Islands. Further investigations are needed on the CTX-producing species, sampling methods, toxin profiles in microalgae and fish, relations between macroalgae, microalgae, and fish, and the accumulation of CTXs throughout the food webs. Until these issues are solved, it will be challenging to implement the best decisions to prevent CP at the local level of the Archipelago.

## 4. Methods

### 4.1. Reagents and Equipment

CTX1B standard solution was obtained from Prof. Richard J. Lewis (The Queensland University, Australia) [60] and calibrated (correction factor of 90%) in relation to the NMR-quantified CTX1B standard solution from Prof. Takeshi Yasumoto (Japan Food Research Laboratories, Japan). 51-OH-CTX3C standard solution was kindly provided by Prof. Takeshi Yasumoto (Japan Food Research Laboratories, Japan) and was used as a model for the series of CTX3C congeners. Neuroblastoma murine cells (neuro-2a) were purchased from ATCC LGC standards (USA). Poly-L-lysine, fetal bovine serum (FBS), L-glutamine solution, ouabain, veratridine, phosphate buffered saline (PBS), penicillin, streptomycin, RPMI-1640 medium, sodium pyruvate, thiazolyl blue tetrazolium bromide (MTT), *N*-(3-dimethylaminopropyl)-*N*′-ethylcarbodiimide hydrochloride (EDC), *N*-Hydroxysuccinimide (NHS), Tween 20, bovine serum album (BSA), poly horseradish peroxidase–streptavidin (polyHRP–streptavidin), and 3,3′,5,5′-tetramethylbenzidine (TMB) liquid substrate were supplied by Merck KGaA (Germany). Dimethyl sulfoxide (DMSO) and absolute methanol were purchased from Honeywell (Spain) and Chemlab (Spain), respectively. Taq Polymerase and Dynabeads M-270 Carboxylic Acid (2 × 109 beads mL^−1^) were purchased from Invitrogen (Spain). QIAquick PCR Purification Kit was obtained from Qiagen (Germany). For the DNA amplification, a Mastercycler nexus gradient thermal cycler purchased from Eppendorf (Spain) was used. The 3G8, 10C9, and 8H4 monoclonal antibodies (mAbs) had been prepared by immunizing mice with keyhole limpet hemocyanin (KLH) conjugates of rationally designed synthetic haptens [79,80,81,82,83,84,85]. A microplate Reader KC4 (BIO-TEK Instruments, Inc., Vermont, VT, USA) was used to perform colorimetric measurements, and Gen5 software was used to collect and process the data. Arrays of eight screen-printed carbon electrodes (DRP-8 × 110), a boxed connector (DRP-CAST8X), and a magnetic support (DRP-MAGNET8X) were purchased from Dropsens S.L. (Spain). A PalmSens potentiostat (PalmSens, Houte, Netherlands) connected to an 8-channel multiplexer (MUX8) was used to perform amperometric measurements. Data from potentiostat were collected and evaluated with PalmSens PC software (PalmSens, Houte, Netherlands).

### 4.2. Sampling Area and the Strategy

The Canary Islands are located in the north-east Atlantic Ocean (28.36715°–17.61396°) between 100 km and 600 km west of the north-west African coast. Samplings were performed in seven big islands during 2016 and 2017. Gran Canaria, Fuerteventura, and Lanzarote were sampled in October 2016 and La Palma, Tenerife, El Hierro, and La Gomera were sampled in October 2017. Additionally, samplings were carried out in Gran Canaria in April 2017. Sampling locations are shown on a map in Figure 5, and Table 4 shows the details. In total, 53 sampling stations were studied. In each station, the temperature, salinity, pH, oxygen saturation, and oxygen concentration were recorded with a multiparametric probe (YSI 556 MPS). The coordinates of the stations, date, and the environmental data are shown in Appendix A. In each sampling station, two samples were obtained: epilithic (surface rasping of rocky substrates) and epiphytic (sampling of macrophytes). Samples were collected using a plastic bottle (Nalgene, HDPE, 1 L). For the first sample, the substrates were scratched using the bottleneck, and for the second, macroalgae were gently removed from their substrate and introduced into the bottle under the water. Then, the bottles were manually shaken, and each sample was filtered through a 300 µm nylon mesh to remove the detritus and the larger grazers. The filtered water was stored in a plastic bottle (Nalgene, HDPE, 125 mL).

### 4.3. Isolation and Culturing

Samples were observed under a light microscope (Leica Microsystems GmbH, Germany) to isolate *Gambierdiscus* and *Fukuyoa* cells by the capillary method [86]. Each dinoflagellate cell was placed individually in one well of an untreated Nunc 24 well plate (Thermo Fisher Scientific) with 1 mL of modified ES medium [87]. The culture medium was constituted with seawater from L’Ametlla de Mar (Spain), Mediterranean Sea (40.8465°; 0.772432°), which was aged for two months in the dark and was filtered through an activated carbon filter of PTFE (Thermo Fisher Scientific) and after through a 0.22 µm cellulose acetate filter (Merck KGaA, Germany). The salinity was adjusted to 36 with Milli-Q water. After 2–3 weeks, fifty-two cultures achieved at least 20 cells mL^−1^ and cells were transferred to fresh medium for maintenance in 28 mL round-bottom glass tubes (Thermo Fisher Scientific). Cells were cultured at 24 ± 1 °C, with a photon irradiance of 100 μmol photons m^−2^ s^−1^ and 12:12 light:dark cycle. Light was provided by fluorescent tubes with white light. Irradiance was measured by QSL-2100 Radiometer (Biospherical Instruments, San Diego, CA, USA).

### 4.4. Molecular Identification

To identify the fifty-two cultures at the species level, the D8-D10 region of the LSU rDNA was used [43,44]. The cultures were transferred to 100 mL Erlenmeyer flasks at a final concentration of 50 cells mL^−1^. Afterwards, 50 mL of culture were harvested at the exponential phase by centrifugation (4300 g, 20 min). The resulting cell pellet was processed for DNA extraction using the phenol/chloroform/isoamylalcohol extraction (PCI) protocol according to Toldrà et al. [88]. Genomic DNA was quantified and checked for its purity using a NanoDrop 2000 spectrophotometer (Thermo Fisher Scientific). The D8-D10 of the LSU region was amplified by PCR using the primers FD8 and RB [43]. Amplifications were carried out in a Mastercycler nexus gradient thermal cycler (Eppendorf, Spain) as it was described in Reverté et al. [61]. The similarity of sequences was checked by BLAST (National Centre of Biotechnology Information, NCBI) and deposited in GenBank. Sequences of >562 were aligned using MAFFT v.7 [89] with the G-INS-1 progressive method. The final alignment consisted of 42 seqs from the current study with 562 positions. The origin of the sequences and the date of collection are shown in Appendix A. The phylogenetic relationships were inferred by Maximum Likelihood (ML) using RaxML v.8 [90] and by Bayesian inference (BI) using Mr. Bayes v.3.2.2 [91]. The model of evolutionary reconstruction for ML analysis was estimated using JModelTest 2.1.10 [92]. In the BI approach, two analyses were run in parallel, 10^6^ generations, and four chains in each run. The parameters used for analysis were nst = mixed and rates = gamma. By default, 25% of the trees were discarded. The stability of the chains was checked using Tracer v.1.7.1 [93].

### 4.5. Morphological Characterization

Monoclonal cultures were cultivated and acclimated for a minimum of one year before experimentation to reduce the variability of the response that stress can produce [94]. The morphological characterization was conducted in 4 strains: IRTA-SMM-16-286 (*G. australes*) from Lanzarote, IRTA-SMM-17-407 (*G. excentricus*) from La Gomera, IRTA-SMM-17-03 (*G. caribaeus*),these three strains were reported in Gaiani et al. [68], and IRTA-SMM-17-421 (*G. belizeanus*) from El Hierro. A 5 mL aliquot of each culture at the late exponential phase was fixed with 3% Lugol’s iodine solution. Then, the aliquots were stained using Calcofluor White M2R (Sigma Aldrich, Spain), according to Fritz and Triemer [95]. The thecae were described following the nomenclature proposed by Fraga et al. [30]. The depth (dorso-ventral axis) and the width (transdiameter lateral extremes of cingulum) of 50 individuals of each strain were measured using an epifluorescence microscope (LEICA DMLB and NIKON eclipse 80i) equipped with an Olympus camera (Olympus DP70). The software used for measurements was an Olympus DP controller (Olympus Corporation). Cell dimensions were expressed as mean ± standard deviation (SD). In addition, cells of *G. belizeanus* (IRTA-SMM-17-421) were observed by scanning electron microscopy (SEM). For that, 10 mL of cultures at the initial exponential growth phase were fixed with glutaraldehyde at a final concentration of 4% for 2 h at room temperature. After that, 3 mL of culture were collected with a syringe by applying a low pressure on a 5 µm Nuclepore Track-Etch Membrane (Thermo Fisher Scientific). Previously, the membrane had been coated with poly-L-lysine and held in a plastic filter mold of 13 mm diameter (PALL, life Science). The membrane with the cells was rinsed twice: once with seawater (autoclaved and filtered with active carbon 0.22 µm) and a second time with seawater/MilliQ water (50:50, v:v). Afterwards, dehydration was performed in a graded EtOH series of 30, 50, 70, 80, 90 and twice with 96%. Filters were transferred to vessels with absolute EtOH and sent to SEM facilities of the Institut de Ciències del Mar (ICM-CSIC). Then, filters were submitted to critical-point drying with liquid carbon dioxide in a BAL-TEC CPD030 unit (Leica Microsystems, Austria). Dried filters were mounted on stubs with colloidal silver and then sputter-coated with gold in a Q150R S (Quorum Technologies Ltd.). Cells were observed with a Hitachi S3500N scanning electron microscope (Hitachi High Technologies Co., Ltd., Japan) at an accelerating voltage of 5 kV.

### 4.6. Production of Microalgal Extracts for Toxin Evaluation

Forty-one cultures of *Gambierdiscus* were inoculated in 500 mL Fernbach at an initial concentration of 50 cells mL^−1^. When cultures arrived at the late exponential-early stationary phase, aliquots of microalgal cultures were collected and fixed with 3% Lugol’s iodine solution for cell counting under the light microscope. Then, cultures were harvested by centrifugation (4300 g, 20 min), obtaining between 1 × 10^5^–1.6 × 10^6^ cells of each strain. Cell pellets were kept with absolute methanol (1 mL of methanol for 1 x 10^−6^ cells) at −20 °C. For toxin extraction, each microalgal pellet was sonicated using an ultrasonic cell disrupter (Watt ultrasonic processor VCX750, USA) at 3 s on and 2 s off, 34% amplitude for 15 min. After that, the sample was centrifuged (600 g, 5 min), and the supernatant was removed and kept in a glass vial. Then, new absolute methanol was added to the cell pellet and the whole process of toxin extraction was repeated twice. Supernatants were pooled and evaporated under N_2_ by Turbovap (Caliper, Hopkinton, MA, USA) at 40 °C. Dried extract was dissolved with methanol and kept at −20 °C until toxin evaluation.

### 4.7. Evaluation of CTX-Like Toxicity with the Neuroblastoma Cell-Based Assay

The CTX-like toxicity of forty-one cultures of *Gambierdiscus*, consisting of *G. australes* (*n* = 29), *G. excentricus* (*n* = 10), *G. caribaeus* (*n* = 1), and *G. belizeanus* (*n* = 1), was evaluated using the neuro-2a CBA. This assay is used to detect compounds that target voltage-gated sodium channels (VGSCs). The assay uses ouabain, which blocks the Na^+^/K^+^-ATPase ion pump and inhibits the efflux of Na^+^ [96] and veratridine, which activates the VGSC, enhancing the influx of Na^+^ [97]. The neuro-2a CBA is based on the reduction of viability of the neuro-2a cells when the extract presents CTXs or molecules that activate VGSCs after the ouabain and veratridine treatment [98].

The neuro-2a CBA and the data analysis were conducted following the protocol described in Caillaud et al. [99]. CTX1B was the molecule of reference (standard) to quantify the CTX-like toxicity of the extracts, and a dose-response curve was obtained each day of experimentation. To be able to discriminate CTX-like toxicity from other types of toxicity, neuro-2a cells were exposed to extracts or standards with and without ouabain and veratridine. It was considered that samples contained CTX-compounds or other compounds that target VGSCs when toxicity was not observed in the O/V^−^ conditions but was observed in the O/V^+^ conditions. In the cases that toxicity was observed in both conditions (O/V^−^ and O/V^+^), it indicates the presence of a non-specific toxic compounds, other that target the VGSCs.

Concentrations of CTX1B ranged between 0.2 and 25 pg mL^−1^, and concentrations of microalgal extracts ranged between 0.3 and 6000 cells equiv. mL^−1^. The concentrations of microalgal extracts were chosen based on the toxic effect observed in the neuro-2a cells from previous screening experiments. After 24 h of exposure, the viability of the neuro-2a cells was assessed by the quantitative colorimetric MTT assay [98]. Data analysis was performed using SigmaPlot software 12.0 (Systat Software Inc., San Jose, CA, USA). Matrix effect was considered when significant toxicity appeared in the neuro-2a cells without O/V^−^. Significant toxicity was described as the inhibition of more than 20% of the cell viability. The normality of the CTX-like toxicities was checked using the Shapiro-Wilk test. Then, a one-way ANOVA was used to test if significant differences in CTX-like toxicities occurred among *G. excentricus* and *G. australes*, and if significant differences occurred between the islands within these species. The statistical test and graphs were performed with R studio [100].

### 4.8. Evaluation of the Presence of Two Series of CTX Congeners (CTX1B and CTX3C) in G. belizeanus with a Colorimetric Immunoassay and an Electrochemical Immunosensor

The use of 3G8 and 10C9 mAbs in the screening of the *Gambierdiscus* extracts allowed the detection of two series of CTXs congeners (CTX1B and CTX3C) thanks to the high affinities of these capture antibodies for their CTX targets. In particular, 3G8 mAb binds to the left wing of CTX1B and 54-deoxyCTX1B [83], and 10C9 mAb binds to the left wing of CTX3C and 51-hydroxyCTX3C [81]. The 8H4 mAb, used as a reporter antibody, binds to the right wing of all the four congeners [101]. For this reason, quantifications are expressed in fg cell^−1^ of CTX1B equiv. when only the 3G8 mAb was incubated with the microalgal extract and in fg cell^−1^ of 51-hydroxyCTX3C equiv. in presence of only the 10C9 mAb. The use of separated mAbs allows the discrimination between the two series of CTX congeners. Therefore, when both antibodies are incubated with the extract, a global response is obtained, and thus quantifications can be provided either in fg cell^−1^ of CTX1B equiv. or 51-hydroxyCTX3C equiv. In this work, the obtained quantifications when the two antibodies are incubated together are provided only in CTX1B equiv. for comparison with neuro-2a CBA results. Analyses of *G. belizeanus* extracts were performed as described in Gaiani et al. [68]. Briefly, magnetic beads (MBs) were activated with an EDC and NHS solution and incubated with 3G8 or 10C9 mAbs. In particular, 3G8 mAb binds against the left wing of CTX1B and 54-deoxyCTX1B. Instead, 10C9 mAb binds specifically to the left wing of CTX3C and 51-hydroxyCTX3C [83,85]. Figure 6 shows the structure of the four CTX congeners (CTX1B and CTX3C congeners) that the antibodies recognize. After the incubation, the mAb–MB conjugates were washed, placed into new tubes in a separate or mixed way, exposed to microalgal extract (previously evaporated and suspended in PBS–Tween) or CTX standard (CTX1B or 51-hydroxyCTX3C) for calibration purposes. Afterwards, a blocking step was performed with PBS–Tween–BSA. Then, the conjugates were incubated with biotin-8H4 mAb [101]. The 8H4 mAb binds to the right wing of CTX1B and 54-deoxyCTX1B and has cross-reactivity with the right wing of CTX3C and 51-hydroxyCTX3C. Finally, immunocomplexes were incubated with polyHRP–streptavidin, washed, and re-suspended in PBS–Tween. The colorimetric immunoassay was performed incubating the immunocomplexes with TMB (HRP enzyme substrate) and reading the absorbance at 620 nm using an automated plate spectrophotometer. Measurements were performed in triplicate. The electrochemical immunosensor was performed placing the immunocomplexes on the working electrodes of an 8-electrode array, incubating with TMB, and measuring the reduction current using amperometry (−0.2 V (vs. Ag) for 5 s). Measurements were performed in quadruplicate.

## Figures and Tables

**Figure 1 toxins-12-00692-f001:**
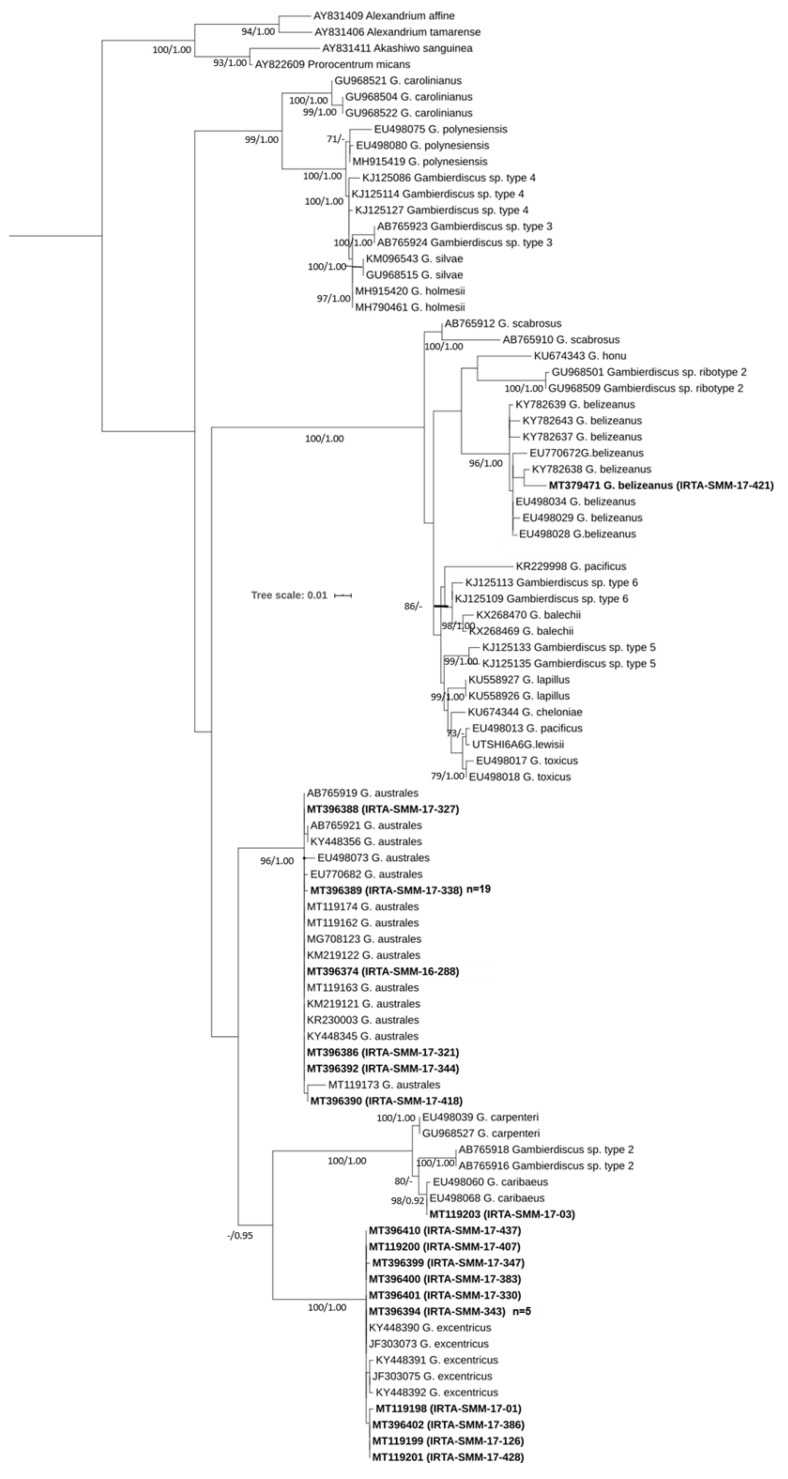
Phylogenetic tree of the LSU D8-D10 region (rDNA) using Maximum Likelihood analysis. Sequences in bold represent the strains of this study. The number of clones (*n*) with the same haplotypes is shown in parentheses. Values at nodes represent bootstrap values (≥70) and the Bayesian posterior probability (≥0.95) (bt/pp).

**Figure 2 toxins-12-00692-f002:**
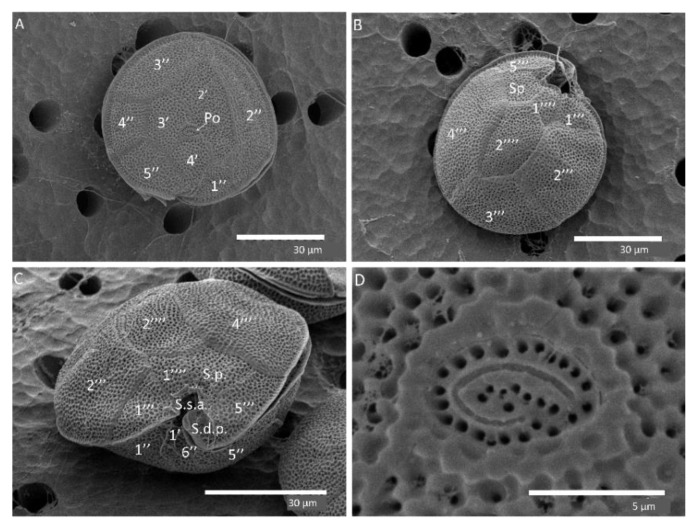
SEM images of *G. belizeanus* (IRTA-SMM-17-421): apical (**A**), antapical (**B**), ventral (**C**) views, detail of Po plate and pores (**D**).

**Figure 3 toxins-12-00692-f003:**
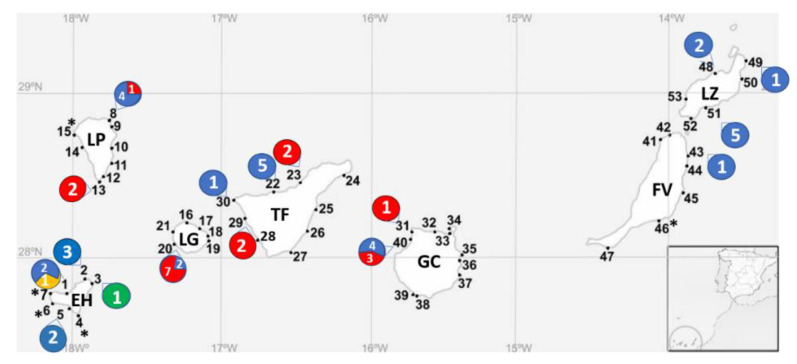
Distribution of each species in the stations of the Canary Islands during 2016–2017. Station numbers are represented in bold. The presence of *Gambierdiscus* species determined with molecular analysis is presented with a circle and includes the number strains identified for each species. The asterisk represents the presence of *Gambierdiscus* sp. Colors of circles are for *G. australes* (blue), *G. excentricus* (red), *G. caribaeus* (green), and *G. belizeanus* (yellow). EH (El Hierro), FV (Fuerteventura), GC (Gran Canaria), LG (La Gomera), LP (La Palma), LZ (Lanzarote), and TF (Tenerife).

**Figure 4 toxins-12-00692-f004:**
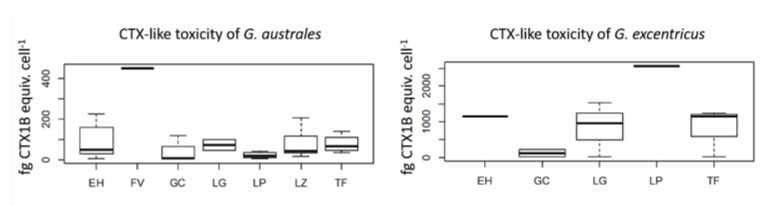
Distribution of CTX-like toxicity of *G. australes* and *G. excentricus* according to island of origin. EH (El Hierro), FV (Fuerteventura), GC (Gran Canaria), LG (La Gomera), LP (La Palma), LZ (Lanzarote) and TF (Tenerife).

**Figure 5 toxins-12-00692-f005:**
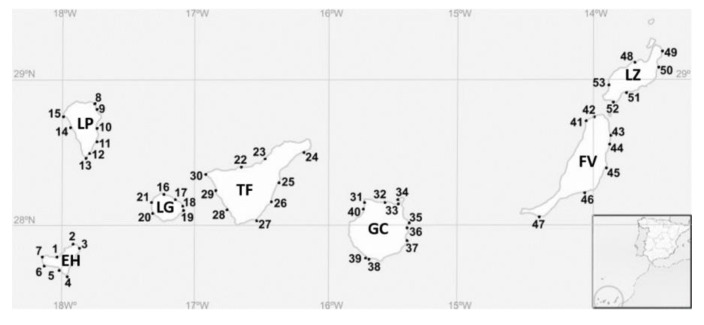
Sampling stations in the Canary Islands for the current study. Numbers correspond to locations described in Table 1. EH (El Hierro), FV (Fuerteventura), GC (Gran Canaria), LG (La Gomera), LP (La Palma), LZ (Lanzarote), and TF (Tenerife).

**Figure 6 toxins-12-00692-f006:**
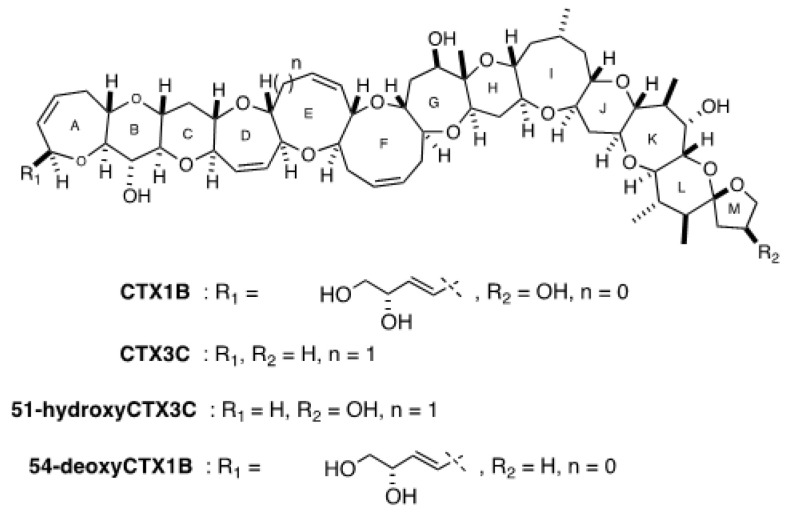
Structure of CTX1B and CTX3C congeners recognized by the antibodies used in this work.

**Table 1 toxins-12-00692-t001:** Morphological sizes average of depth and width (±SD) of *Gambierdiscus* species of this study measured with light microscopy. The ranges of values are shown in parentheses.

Species	Depth (µm)	Width (µm)
*G. australes*	71.13 ± 7.06 (60.6–98.4)	65.40 ± 6.54 (53.4–82.1)
*G. belizeanus*	64.12 ± 5.28 (52.8–76.2)	59.64 ± 5.95 (46.5–76.0)
*G. caribaeus*	87.20 ± 11.19 (61.2–116.5)	86.45 ± 11.44 (63.17–119.7)
*G. excentricus*	90.25 ± 8.90 (72.2–109.7)	82.97 ± 9.06 (67.8–106.7)

**Table 2 toxins-12-00692-t002:** Literature review of the distribution of the species in the Canary Islands. EH (El Hierro), FV (Fuerteventura), GC (Gran Canaria), LG (La Gomera), LP (La Palma), LZ (Lanzarote), and TF (Tenerife).

Species	Islands	Date of Sampling (References)
*G. australes*	LZ, FV, GC	September–October 2015 [33] October 2016 (this study)
	EH	September–October 2015 [33] October 2017 (this study)
	TF	2013 [31] September–October 2015 [33] October 2017 (this study)
	LG	October 2017 ([33] and this study)
	LP	October 2017 (this study)
*G. belizeanus*	EH	October 2017 (This study)
*G. caribaeus*	EH	September–October 2015 [33] October 2015 [55] October 2017 (this study)
	LG	October 2017 [32]
*G. carolinianus*	TF	September–October 2015 [33]
*G. excentricus*	TF	March 2004 [30] 2013 [31] September–October 2015 [33] October 2017 (This study)
	LZ	September–October 2015 [33]
	FV	September–October 2015 [33]
	GC	September–October 2015 [33] April 2017, October 2017 (this study)
	LP	March 2004 [30] October 2017 ([33] and this study)
	LG	March 2004 [30] October 2017 ([33] and this study)
*G. silvae*	TF	2013 [31] September–October 2015 [33]
	GC	Winter 2010 [31]
	LG	October 2017 [32]

**Table 3 toxins-12-00692-t003:** Ciguatoxin (CTX)-like toxicity of *Gambierdiscus* spp. using the neuro-2a CBA. CTX-like toxicity and the limit of detection (LOD) are expressed as fg CTX1B equiv. cell^−1^. Ref: number of the station according to Figure 1, EH (El Hierro), FV (Fuerteventura), GC (Gran Canaria), LG (La Gomera), LP (La Palma), LZ (Lanzarote) and TF (Tenerife).

Strain Code	CTX-Like Toxicity	Island	Station	Strain Code	CTX-Like Toxicity	Island	Station
*G. australes*				*G. australes*			
IRTA-SMM-17-004	205.1 ± 34.5	LZ	48	IRTA-SMM-17-316	51.5 ± 6.9	TF	22
IRTA-SMM-17-006	127.7 ± 85.0	LZ	51	IRTA-SMM-17-307	37.3 ± 12.6	TF	30
IRTA-SMM-16-288	106.1 ± 75.3	LZ	51	IRTA-SMM-17-436	98.6 ± 25.4	LG	20
IRTA-SMM-16-290	46.1 ± 22.2	LZ	51	IRTA-SMM-17-393	44.6 ± 11.5	LG	20
IRTA-SMM-16-292	39.7 ± 10.5	LZ	49	IRTA-SMM-17-344	41.2 ± 0.1	LP	8
IRTA-SMM-16-286	33.6 ± 6.5	LZ	51	IRTA-SMM-17-335	29.1 ± 8.6	LP	8
IRTA-SMM-16-293	32.7 ± 10.0	LZ	51	IRTA-SMM-17-287	11.3 ± 2.3	LP	8
IRTA-SMM-17-007	15.8 ± 1.7	LZ	48	IRTA-SMM-17-288	5.7 ± 3.8	LP	8
IRTA-SMM-17-002	452.6 ± 23.2	FV	43	IRTA-SMM-17-389	226.3 ± 24.5	EH	1
IRTA-SMM-17-103	118.7 ± 30.3	GC	40	IRTA-SMM-17-324	160.4 ± 17.2	EH	5
IRTA-SMM-17-107	12.2 ± 2.1	GC	40	IRTA-SMM-17-418	68.3 ± 9.5	EH	2
IRTA-SMM-17-112	1.9 ± 0.6	GC	40	IRTA-SMM-17-321	31.9 ± 15.0	EH	5
IRTA-SMM-17-106	1.7 ± 0.1	GC	40	IRTA-SMM-17-425	27.8 ± 3.3	EH	2
IRTA-SMM-17-358	138.9 ± 17.7	TF	22	IRTA-SMM-17-327	7.2 ± 0.3	EH	2
IRTA-SMM-17-291	82.8 ± 22.2	TF	22				
*G. belizeanus*				*G. caribaeus*			
IRTA-SMM-17-421	5.6 ± 0.1	EH	1	IRTA-SMM-17-003	Neg. LOD < 0.42	EH	3
*G. excentricus*				*G. excentricus*			
IRTA-SMM-17-001	1149.3 ± 212.3	EH	1	IRTA-SMM-17-386	12.8 ± 2.8	TF	23
IRTA-SMM-17-126	226.7 ± 22.1	GC	40	IRTA-SMM-17-429	1525.9 ± 634.1	LG	20
IRTA-SMM-17-128	9.5 ± 2.6	GC	40	IRTA-SMM-17-432	962.1 ± 154.7	LG	20
IRTA-SMM-17-404	1257.64.8 ± 319.3	TF	29	IRTA-SMM-17-413	18.1 ± 5.7	LG	20
IRTA-SMM-17-405	1153.4 ± 238.8	TF	29	IRTA-SMM-17-330	2566.7 ± 333.3	LP	13

**Table 4 toxins-12-00692-t004:** Description of the sampling stations of the present study in the Canary Islands. Ref.: correspond to the locations in Figure 1. Details of the date, coordinates, and environmental data are compiled in Appendix A.

Ref.	Island	Location	Ref.	Island	Location
1	El Hierro	Charco Azul	27	Tenerife	La Tejita
2	El Hierro	Charco Manso	28	Tenerife	La Caleta
3	El Hierro	Tamaduste	29	Tenerife	El Pto. de Santiago
4	El Hierro	La Restinga	30	Tenerife	Punta de Teno
5	El Hierro	Tacoron	31	Gran Canaria	Punta Sardina
6	El Hierro	Orchilla	32	Gran Canaria	El Puertillo, Bañaderos
7	El Hierro	Verodal	33	Gran Canaria	Las Canteras
8	La Palma	La Fajana	34	Gran Canaria	El Confital
9	La Palma	El Puerto Espíndola	35	Gran Canaria	Melenara
10	La Palma	Los Cancajos	36	Gran Canaria	Playa Tufia
11	La Palma	Salemera	37	Gran Canaria	Agüimes, Playa El Cabrón
12	La Palma	El Puerto de Trigo	38	Gran Canaria	Arguineguín El Pajar
13	La Palma	El Faro Fuencaliente	39	Gran Canaria	Arguineguín Sta. Águeda
14	La Palma	Tazacorte	40	Gran Canaria	Las Charcas de Agaete
15	La Palma	Puntagorda	41	Fuerteventura	Caleta del Río, El Cotillo
16	La Gomera	Vallehermoso	42	Fuerteventura	Majanicho
17	La Gomera	La Caleta	43	Fuerteventura	Playa Jabalito
18	La Gomera	Playa de Ávalos	44	Fuerteventura	Puerto Lajas
19	La Gomera	Playa de la Cueva	45	Fuerteventura	Puerto Caleta del Fuste
20	La Gomera	Playa de Vueltas	46	Fuerteventura	Gran Tarajal
21	La Gomera	Alojera	47	Fuerteventura	Morro Jable
22	Tenerife	Charca del Viento	48	Lanzarote	Caleta Caballo
23	Tenerife	Puerto del Sauzal	49	Lanzarote	Las Cocinitas
24	Tenerife	Playa las Teresitas	50	Lanzarote	Charco del Palo
25	Tenerife	El Puertito	51	Lanzarote	Puerto Calero
26	Tenerife	Punta de Abona	52	Lanzarote	Playa Mujeres
			53	Lanzarote	El Golfo

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
