# Peer review of "Further Advance of Gambierdiscus Species in the Canary Islands, with the First Report of Gambierdiscus belizeanus"

_toxins, 2020, doi:10.3390/toxins12110692_

Round 1

Reviewer 1 Report

General comments:

  1. The main focal point of this manuscript is that CTX-1B activity was observed in algal extracts. CTX-1B is regarded as a fish metabolite that is produced by the spiroisomerisation and oxidation of CTX-4A, the algal precursor. It is the reviewer’s opinion reporting that algal extracts contain CTX-1B is misleading. If such a claim is to be made then targeted analysis, using for example LC-MS/MS, should be performed to confirm the presence of this analogue.
  2. Fukuyoa has not been demonstrated to produce any ciguatoxins or maitotoxins. This should be omitted from the manuscript.

Line 33: There are more species than just fish and shellfish that have been reported to cause CP. Refer FAO document on the 2018 expert meeting on CP.

Line 135: ‘twenty-one’ should be numerical, 21.

Lines 166-167: Sentence begins with “The present study…”, the next sentence begins with “In the current study…”. These are essentially the same comment so should be adjusted.

Lines 193-194: Two consecutive sentences start with “Gambierdiscus species…”. These could be combined as a single sentence.

Line 226: 21.8 to 27.9 °C; Line 230: 26.1-29.1 °C and 25-31 °C. Reporting of the temperature range should be consistent.

Line 271: The toxicity of G. australes is a result of MTX-1 production, which has been demonstrated to have untargeted toxicity to neuro-2a cells. There are no confirmed reports of this species producing CTXs.

Lines 293-295: The authors state the other compounds present in the methanolic extract can cause toxicity. Yet in the present study there is no consideration for this. Why not?

Lines 359-360: The authors refer to specific binding on CTXS-1B, 54-deoxyCTX-1B and 51-hydroxyCTX-3C. These are all known fish metabolites and are not produced by micro algae. Having these analogues referenced in this manner is misleading for any readers of this manuscript.

Line 404: Fukuyoa has not been demonstrated to contribute to CP. Analysis of this species has been shown to be of low-no toxicity. Refer Munday et al 2017.

Lines 405-407: The authors reference MTX-3. This compound has been reclassified as 44-methylgambierone, as per the references used. This should be updated in the text.

Lines 405-406: 44-methylgambierone has been demonstrated to be of low toxicity, stating that it is the same as CTX-3C is misleading.

Line 483: No Fukuyoa cells were identified so there is no need for this genus to be included here.

Line 598: It is stated a 51-hydroxyCTX3C standard is used, however it is not included in section 3.1 – Reagents and equipment.

Author Response

General comments:

  1. The main focal point of this manuscript is that CTX-1B activity was observed in algal extracts. CTX-1B is regarded as a fish metabolite that is produced by the spiroisomerisation and oxidation of CTX-4A, the algal precursor. It is the reviewer’s opinion reporting that algal extracts contain CTX-1B is misleading. If such a claim is to be made then targeted analysis, using for example LC-MS/MS, should be performed to confirm the presence of this analogue.

 Our manuscript does not intend to define unequivocally the CTX analogues potentially present since this objective definitively would need the use of LC-MS/MS or LC-HRMS. Our detection method is based on a cell-based assay (CBA), and we detect toxicity, not a specific molecule. As for any toxicological study, the evaluation of the toxicity has to be reported according to a given standard, and, in a systematic approach, our laboratory uses CTX-1B as the reference molecule in order to calibrate the toxicity assay, and be able to report the toxicity of microalgae in CTX-1B equivalents. This is a universal criterion used by everyone working on functional assays such as CBA, with flexibility to select the most appropriate CTX congener, such as CTX-1B, CTX3C or other.

 In our manuscript, we always express the toxicity as CTX-like toxicity or CTX-like activity, which can be composite toxicity and we do not say that the toxicity is due to CTX-1B, this it was mentioned in old lines: 293-295. Note that, to express toxicity, we always use the word of equivalents.

To avoid confusions, we have changed CTX-like activity by CTX-like toxicity everywhere. Although LC-MS/MS is a useful technique to identify which CTX congeners are present in a sample, the low amount of toxin found in the microalgae and the lower limit of detection of LC-MS/MS compared to CBA does not allow CTX congeners identification.

  1. Fukuyoa has not been demonstrated to produce any ciguatoxins or maitotoxins. This should be omitted from the manuscript.

Yes, we could omit the Fukuyoa in the manuscript because it has never been found in the Canary Islands. Fukuyoa has been demonstrated to exhibit CTX-like compounds (Litaker et al., 2017). In addition, Roeder et al. (2010) mentioned that they reported a peak of CTX analogue by LC-MS/MS for the isolate NOAA3 in G. ruetzleri strain, which presently is Fukuyoa ruetzleri (included in the Fukuyoa genus). Moreover Laza-Martínez et al. (2016) has confirmed the presence of trace amounts of compounds having mass characteristics of 54-deoxyCTX1B by LC-HRMS.

Roeder, K.; Erler, K.; Kibler, S.; Tester, P.; Van The, H.; Nguyen-Ngoc, L.; Gerdts, G.; Luckas, B. Characteristic profiles of Ciguatera toxins in different strains of Gambierdiscus spp. Toxicon 2010, 56, 731–738.

Litaker, R.W.; Holland, W.C.; Hardison, D.R.; Pisapia, F.; Hess, P.; Kibler, S.R.; Tester, P.A. Ciguatoxicity of Gambierdiscus and Fukuyoa species from the Caribbean and Gulf of Mexico. PLoS One 2017, 12, 1–19.

Laza-Martínez, A.; David, H.; Riobó, P.; Miguel, I.; Orive, E. Characterization of a Strain of Fukuyoa paulensis (Dinophyceae) from the Western Mediterranean Sea. J. Eukaryot. Microbiol. 2016, 63, 481–497.

Line 33: There are more species than just fish and shellfish that have been reported to cause CP. Refer FAO document on the 2018 expert meeting on CP.

We agree with reviewer’s suggestion. We have included this reference and we have changed the line: 34 and also in the abstract line 6. Now it reads: “Humans can get poisoned after the consumption of CTX-contaminated fish or very rarely some invertebrates (crustaceans, echinoderms and bivalves), and suffer a disease known as Ciguatera Poisoning (CP)”.

Line 135: ‘twenty-one’ should be numerical, 21.

We have changed twenty-one by 21. (line 154).

Lines 166-167: Sentence begins with “The present study…”, the next sentence begins with “In the current study…”. These are essentially the same comment so should be adjusted.

To avoid repetitions, the first sentence has been deleted (lines: 187-189).

Lines 193-194: Two consecutive sentences start with “Gambierdiscus species…”. These could be combined as a single sentence.

We agree with the reviewer’s comment and text have been changed accordingly. Now in lines 200-203. It reads, “Gambierdiscus species have distinct lower and upper thermal limits and optimal temperatures for their growth. Additionally, they are considered to have a high intraspecific variable response in growth depending on the temperature, salinity and irradiance.”

Line 226: 21.8 to 27.9 °C; Line 230: 26.1-29.1 °C and 25-31 °C. Reporting of the temperature range should be consistent.

Temperature ranges have been changed to be consistent in the whole manuscript. It reads: line 234, 21.8-27.9 °C; Line 238:: 26.1-29.1 °C and 25-31 °C.

Line 271: The toxicity of G. australes is a result of MTX-1 production, which has been demonstrated to have untargeted toxicity to neuro-2a cells. There are no confirmed reports of this species producing CTXs.

As it has mentioned in the first comment the toxicological response is measured as quantities of CTX1B equivalents/cell. Our CBA has the appropriate controls (with ouabain and veratridine) to be able to discriminate the CTX-like toxicity. We compare the cell toxicity in both conditions (O/V- and O/V+). Toxicity in O/V- will indicate indicates the presence of a non-specific toxic compound other than a sodium channel activator. We agree that could be MTX-1, although potency of MTX1 is very powerful, and in both treatments (O/V+ and O/V-) we may observe significant toxicity in neuro-2a cells. Maybe at very low concentrations of MTX, we could observe some effect significant effect on O/V+ and none effect on O/V-. But the maitotoxin effect will be expressed as quantities of CTX-1B equivalents/cell.

In the paper, Estevez et al., (2020), we analysed 4 strains of G. australes from the Balearic Islands by LC-MS/MS and none of strains contained MTX1 with LOD and LOQ for the MTX1 MRM transition 482 [M−2H]2−/[M−2H]2− of 0.32 and 0.97 μg mL−1, respectively. But we found 44-methylgambierone and a putative gambierone analogue. To state that the toxicity is due to MTX1 is risky and confirmation by LC-MS/MS methods would be needed to confirm this. In addition, in Roeder et al. 2010 for the isolate CCMP 1653 which is G. australes strain a peak for 2,3-dihydroxyP-CTX-3C was found by LC-MS/MS.

Estevez, P.; Sibat, M.; Leao, J.M.; Tudó, A.; Rambla-Alegre, M.; Aligizak, K.; Gago-Martinez, A.; Diogène, J.; Hess, P. Use of Mass Spectrometry to determine the Diversity of Toxins Produced by Gambierdiscus and Fukuyoa Species from Balearic Islands and Crete (Mediterranean Sea) and the Canary Islands (Northeast Atlantic). Toxins (Basel). 2020, 1–22.

Roeder, K.; Erler, K.; Kibler, S.; Tester, P.; Van The, H.; Nguyen-Ngoc, L.; Gerdts, G.; Luckas, B. Characteristic profiles of Ciguatera toxins in different strains of Gambierdiscus spp. Toxicon 2010, 56, 731–738.

Lines 293-295: The authors state the other compounds present in the methanolic extract can cause toxicity. Yet in the present study, there is no consideration for this. Why not?

As previously mentioned, the toxicity from other compounds has been taken into account by using ouabain and veratridine in the assay. However, high amounts of these compounds could mask the presence of CTXs. This has been clarified in the manuscript, which now reads in lines 302-311:

“In the evaluation of CTX-like toxicity by neuro-2a CBA of the crude extracts, other compounds can interfere, since the evaluated toxicity in neuro-2a CBA is a composite effect, and sometimes other compounds can cause unspecific mortality [25]. In Estevez et al. [62], the toxic effect on neuro-2a cells of the methanolic crude extract of the G. excentricus (IRTA-SMM-17-407) was very high, making the identification/quantification of CTX-like toxicity impossible. This effect was likely due to the presence of MTX4, also was described in  the same paper, since MTXs have high toxicity to neuro-2a cells [63]. In neuro-2a CBA, the use of controls with and without ouabain and veratridine allow the discrimination of CTX-like toxicity. Nevertheless, high amounts of MTXs, for example, and eventually other compounds, could also mask the presence of CTXs”.

Lines 359-360: The authors refer to specific binding on CTXS-1B, 54-deoxyCTX-1B and 51-hydroxyCTX-3C. These are all known fish metabolites and are not produced by micro algae. Having these analogues referenced in this manner is misleading for any readers of this manuscript.

As it is mentioned in the title of section 2.5, and inside that section, the antibodies are detecting structural equivalents of CTX1B and 51-hydroxyCTX-3C (toxins used as models as for calibration purposes). Although immunosensing techniques are not confirmatory, they are useful for screening. Immunosensing results do not mean to indicate the presence of these toxins, but the presence of compounds structurally-related to the families of these toxins (series of CTX congeners). For clarification, a sentence has been added in section 2.5, which now reads (lines 368-373): “The role of the antibodies in the immunosensing tool is not to confirm the presence of individual CTXs congeners as in instrumental analysis techniques, but to screen the presence of compounds with wings structurally similar to those of the four CTX targets (CTX1B, 54-deoxyCTX1B, CTX3C and 51-hydroxyCTX3C). Therefore, and because of the sandwich format of the assay, the analysis is indicating the presence of compounds structurally related to two series of CTXs congeners, although no evidence can be obtained about which specific CTXs congeners are present.

Line 404: Fukuyoa has not been demonstrated to contribute to CP. Analysis of this species has been shown to be of low-no toxicity. Refer Munday et al 2017.

We agree with the reviewer. Now the sentence reads (line 416): “It is not clear whether only the genus Gambierdiscus contributes to CP”.

Lines 405-407: The authors reference MTX-3. This compound has been reclassified as 44-methylgambierone, as per the references used. This should be updated in the text.

We agree with the reviewer. We have updated the text accordingly.

Lines 405-406: 44-methylgambierone has been demonstrated to be of low toxicity, stating that it is the same as CTX-3C is misleading.

Yes, we agree with the reviewer, our explanation was not clear. We have changed the text (lines: 417-421) and now it reads: “For example, Coolia tropicalis produces 44-methylgambierone (previously reported as MTX3), which induces current sodium in human neuroblastoma cells as CTX3C but with lower potency; additionally, neither of these toxins induce cell death in human cortical neurons when exposed at a 20 nM concentration”.

Line 483: No Fukuyoa cells were identified so there is no need for this genus to be included here.

We have deleted it.

Line 598: It is stated a 51-hydroxyCTX3C standard is used, however it is not included in section 3.1 – Reagents and equipment.

The following information has been added to the manuscript (lines:454-458),: CTX1B standard solution was obtained from Prof. Richard J. Lewis (The Queensland University, Australia) and calibrated (correction factor of 90%) in relation to the NMR-quantified CTX1B standard solution from Prof. Takeshi Yasumoto (Japan Food Research Laboratories, Japan). 51-OH-CTX3C standard solution was kindly provided by Prof. Takeshi Yasumoto (Japan Food Research Laboratories, Japan) and was used as a model for the series of CTX3C congeners.

Reviewer 2 Report

The manuscript Toxins-958273 describes Gambierdiscus spp. found in the Canary Islands together with their CTX-like toxicities and toxin production of G. belizeanus which was reported for the first time in this region. In general, it is a well-prepared work providing sufficient baseline data. These findings support that Gambierdiscus spp. are emerging threats to health of marine ecosystems and human, regular surveys of these toxic benthic dinoflagellates and more studies on their toxins are necessary at hotspots of Gambierdiscus spp. like Canary Islands and greater areas especially in a warmer climate.

Some corrections to text editing are needed. My suggestion is Minor Revision.

One major concern:

Please rearrange the section of Results and Discussion, put some background information to Introduction or put introduction of some methods to section of Method. The current form is not clear enough, and it is difficult for readers  to focus on your key points of view in the discussion.

e.g., Line 179-189 about Canary Islands, Line 190-205 about distribution of Gambierdiscus spp., L357-368 about use of 3G8 and 10C9 mAbs, etc.

Should mention major findings (not long) in your study, relevant supporting data from others and your ideas/conclusions (highlights) only in Discussion.  

Other minor comments:

1. Please provide some more background information of G. belizeanus in the Introduction, e.g., the global distribution of this species.

2.G. belizeanus“ in subtitle of 2.2.1 should be in italics.

3. L349 “G excentricus” should be “G. excentricus

4. Inconsistent font style in section 3.1.

5. L470 “1L” should be “1 L”, L473 “125mL” should be “125 mL”

6. L473 “200 μm nylon mesh”. If my understanding is correct, you concentrated algal samples from 1 L to 125 mL through collecting the algal cells on the mesh. Is it ok to use 200 μm nylon mesh to collect Gambierdiscus cells?

7. L483 “identify Gambierdiscus and Fukuyoa”, better to use “isolate” here, i.e., “Samples were observed under a light microscope (Leica Microsystems GmbH, Germany) to isolate Gambierdiscus and Fukuyoa cells by the capillary method [49].”

8. There are many routine methods in your study, please describe those methods briefly, e.g.  molecular identification and neuroblastoma cell-based assay.

9. L573 “40.000” is not clear.

10. Section 2.6 on future research strategies is good. I would like to stress that the toxin profiles and their changes in composition and amount based on instrumental analysis could be very helpful and important to better evaluate risks of Gambierdiscus spp. in future studies. You may wish to mention this in the Conclusion as well.

11. L675 “[8].” A typo?

12. The reference is too long with more than 100 items in a regular research paper. Try to cite the most important papers only and make the structure of your MS more appropriate.

That’s all, thanks. Good luck!

Author Response

Reviewer 2

Comments and Suggestions for Authors

The manuscript Toxins-958273 describes Gambierdiscus spp. found in the Canary Islands together with their CTX-like toxicities and toxin production of G. belizeanus which was reported for the first time in this region. In general, it is a well-prepared work providing sufficient baseline data. These findings support that Gambierdiscus spp. are emerging threats to health of marine ecosystems and human, regular surveys of these toxic benthic dinoflagellates and more studies on their toxins are necessary at hotspots of Gambierdiscus spp. like Canary Islands and greater areas especially in a warmer climate.

Some corrections to text editing are needed. My suggestion is Minor Revision.

One major concern:

Please rearrange the section of Results and Discussion, put some background information to Introduction or put introduction of some methods to section of Method. The current form is not clear enough, and it is difficult for readers to focus on your key points of view in the discussion.

e.g., Line 179-189 about Canary Islands, Line 190-205 about distribution of Gambierdiscus spp., L357-368 about use of 3G8 and 10C9 mAbs, etc.

Should mention major findings (not long) in your study, relevant supporting data from others and your ideas/conclusions (highlights) only in Discussion.  

We have shortened the paragraph 179-189 and we have moved it to the introduction (lines: 60-69). We have kept paragraph 190-205 in the discussion. We have deleted the old lines 300-302, because it was mentioned in the introduction. Besides, we have moved the paragraph 357-368 to the methods (lines 610-621).

Other minor comments:

  1. Please provide some more background information of G. belizeanusin the Introduction, e.g., the global distribution of this species.

We have added the global distribution of the species and the toxicity in the introduction (lines: 77-81)

  1. belizeanus“ in subtitle of 2.2.1 should be in italics.

We have changed it.

  1. L349 “G excentricus” should be “ excentricus

We have changed it.

  1. Inconsistent font style in section 3.1.

We have changed it.

  1. L470 “1L” should be “1 L”, L473 “125mL” should be “125 mL”

We have changed it.

  1. L473 “200 μm nylon mesh”. If my understanding is correct, you concentrated algal samples from 1 L to 125 mL through collecting the algal cells on the mesh. Is it ok to use 200 μm nylon mesh to collect Gambierdiscuscells?

Definitively there was a mistake. We used a 300 μm nylon mesh to remove detritus and larger grazers, allowing Gambierdiscus cells to pass through the net (The maximum diameter in the genus Gambierdiscus described is arround 140 μm). Definitively, a 200 μm mesh could not be appropriate. We have corrected in the text, line 493. We thank the reviewer for this precision.

  1. L483 “identify Gambierdiscusand Fukuyoa”, better to use “isolate” here, i.e., “Samples were observed under a light microscope (Leica Microsystems GmbH, Germany) to isolate Gambierdiscus and Fukuyoa cells by the capillary method [49].”

We have changed it (lines: 503-505).

  1. There are many routine methods in your study, please describe those methods briefly, e.g.  molecular identification and neuroblastoma cell-based assay.

We have reduced the text in the methods and cited the appropriate references for the molecular identification and the neuro-2a cell-based assay.

  1. L573 “40.000” is not clear.

We have deleted this information according to the previous comment. We have referenced the neuro-2a CBA.

  1. Section 2.6 on future research strategies is good. I would like to stress that the toxin profiles and their changes in composition and amount based on instrumental analysis could be very helpful and important to better evaluate risks of Gambierdiscus spp. in future studies. You may wish to mention this in the Conclusion as well.

We have added to the lines 648-650.” Further investigations are needed on the CTX-producing species, sampling methods, toxin profiles in microalgae and fish, relations between macroalgae, microalgae and fish, and accumulation of CTXs throughout the food webs.”

  1. 11. L675 “[8].” A typo?

We have change it.

  1. The reference is too long with more than 100 items in a regular research paper. Try to cite the most important papers only and make the structure of your MS more appropriate.

We have deleted some references. However, we still have almost 100 references because we have added some related with the distribution of G. belizeanus. We think that is appropriate to maintain them because our article includes very different information (geographical distribution, seven species, data for toxicity, immunosensing data, morphology, and molecular identification).

That’s all, thanks. Good luck!

Reviewer 3 Report

The paper describes the genetics, morphology, distribution and ciguatoxin production by Gambierdiscus species. While the genetics, morphology and distribution appear to be appropriately determined, interpretation of ciguatoxicity is flawed, leaving the paper as mostly descriptive or confirmatory.  The determination of ciguatoxin production needs to have appropriate controls to exclude MTX as the cause of cytotoxicity. Specifically, enrichment for ciguatoxins needs to be undertaken and all extracts need to show they are inactive both in the absence of ouabain or in the presence ouabain/veratridine/TTX (10-6M). As the extract tested was crude and not enriched for CTX, all activity detected is likely attributable to maitotoxin. Further, Pacific CTX1B and CTX3C have not been found in contaminated Atlantic fish and Caribbean CTX remain to be shown to interact with the antibody assay chosen, which is unlikely given the differences in their chemistry. Thus the antibody test generated false positive results and cannot be trusted.

Author Response

Reviewer 3

Comments and Suggestions for Authors

The paper describes the genetics, morphology, distribution and ciguatoxin production by Gambierdiscus species. While the genetics, morphology and distribution appear to be appropriately determined, interpretation of ciguatoxicity is flawed, leaving the paper as mostly descriptive or confirmatory.  The determination of ciguatoxin production needs to have appropriate controls to exclude MTX as the cause of cytotoxicity. Specifically, enrichment for ciguatoxins needs to be undertaken and all extracts need to show they are inactive both in the absence of ouabain or in the presence ouabain/veratridine/TTX (10-6M). As the extract tested was crude and not enriched for CTX, all activity detected is likely attributable to maitotoxin.

We agree with the reviewer that some toxic effect can be attributed to maitotoxins also. In neither case we say that the toxicity is from the CTX compounds. As previously mentioned, ouabain and veratridine have been used to discriminate the presence of CTX-like toxicity. In the neuro-2a assay we compare the toxicity in two conditions (O/V+ and O/V-). The extract and the toxicity of the standard, and when this toxicity is significant with the treatment O/V+ but it is not significant in the O/V-, we say that detect CTX-like toxicity. This would not be a problem in terms of giving a representation of the overall ‘toxicity’ of a Gambierdiscus strain. We agree that for confirmatory we need the analytical methods. This has been also clarified in the manuscript thanks to our previous comment to reviewer 1. However, the additional presence of MTXs cannot be discarded.

Further, Pacific CTX1B and CTX3C have not been found in contaminated Atlantic fish and Caribbean CTX remain to be shown to interact with the antibody assay chosen, which is unlikely given the differences in their chemistry. Thus the antibody test generated false positive results and cannot be trusted.

As in our previous response, the antibodies are detecting structural equivalents of CTX1B and 51-hydroxyCTX-3C. Immunosensing results do not mean to indicate the presence of these toxins, usually found in fish, but the presence of compounds structurally-related to the families of these toxins. The sentence added in section 2.5 clarifies this point. Additionally, Pacific ciguatoxins have been found in contaminated fish from Madeira Archipelago (Atlantic Ocean) by Otero et al. (2010).

Otero, P., Pérez, S., Alfonso, A., Vale, C., Rodríguez, P., Gouveia, N.N., Gouveia, N., Delgado, J., Vale, P., Hirama, M., Ishihara, Y., Molgó, J., Botana, L.M., 2010. First toxin profile of ciguateric fish in Madeira Arquipelago (Europe). Anal. Chem. 82, 6032–6039. https://doi.org/10.1021/ac100516q

Round 2

Reviewer 1 Report

The authors have provided good responses to the reviewer comments and this is appreciated.

However, major fundamental flaws in the manuscript are still present, which is not to be taken personally by the scientists. 

The authors are using Pacific ciguatoxins in the assays to determine toxicity, when the samples are collected from the Caribbean. The Caribbean ciguatoxins are structurally different to the Pacific analogues and have different binding affinities. The authors should be using C-CTX-1, 2, 3 or 4. Reporting toxicity based on Pacific ciguatoxins leads to confusion for new researchers to this field. 

Reporting CTX-1B toxicity/acitivity from microalgae is misleading and should not be published. This is a fish metabolite. 

Further to this using fish metabolite reference material (CTX-1B and 51-hydroxyCTX-3C) to determine toxicity of microalgae is flawed. Toxicity should be referenced to the microalgae metabolites, for example CTX-3B/C or CTX-4A/B. It is well documented that the hydroxylation of the microalgae analogues, to generate the fish metabolites, dramatically increases the toxicity of these compounds.

Refer Yogi et al 2014 "Determination of toxins involved in ciguatera fish poisoning in the Pacific by LC/MS".

The report of Fukuyoa producing 54-deoxyCTX-1B has been questioned by many scientists. Once again this is a fish metabolite so is not produced by microalgae.

Refer to Ikehara et al 2017 "Biooxidation of ciguatoxins leads to species-specific toxin profiles".

The authors mention they monitored MTX-1 using the MRM transition m/z 482 that represents [M-2H]2->[M-2H]2-. This is incorrect. The mass of MTX-1 is 3,425.9 Da as the disodium salt. The doubly charged pseudoMRM to monitor is m/z 1689.6.

Author Response

REVIEWER 1.

Comments and Suggestions for Authors

The authors have provided good responses to the reviewer comments and this is appreciated.

However, major fundamental flaws in the manuscript are still present, which is not to be taken personally by the scientists. 

The authors are using Pacific ciguatoxins in the assays to determine toxicity, when the samples are collected from the Caribbean. The Caribbean ciguatoxins are structurally different to the Pacific analogues and have different binding affinities. The authors should be using C-CTX-1, 2, 3 or 4. Reporting toxicity based on Pacific ciguatoxins leads to confusion for new researchers to this field. 

Reporting CTX-1B toxicity/acitivity from microalgae is misleading and should not be published. This is a fish metabolite. 

Further to this using fish metabolite reference material (CTX-1B and 51-hydroxyCTX-3C) to determine toxicity of microalgae is flawed. Toxicity should be referenced to the microalgae metabolites, for example CTX-3B/C or CTX-4A/B. It is well documented that the hydroxylation of the microalgae analogues, to generate the fish metabolites, dramatically increases the toxicity of these compounds.

Refer Yogi et al 2014 "Determination of toxins involved in ciguatera fish poisoning in the Pacific by LC/MS".

The report of Fukuyoa producing 54-deoxyCTX-1B has been questioned by many scientists. Once again this is a fish metabolite so is not produced by microalgae.

Refer to Ikehara et al 2017 "Biooxidation of ciguatoxins leads to species-specific toxin profiles".

Samples in our study are from the North East Atlantic, not the Caribbean. The use of a standard in toxicological studies is arbitrary, and in our laboratory, we use CTX1B as a reference for all evaluations (Over more than 20 papers published) regardless of the origin of the sample. In toxicology, when certified standards are not available, one reference molecule can be used in order to estimate the toxicity of the samples. What is important is to evaluate the toxicity of the samples according to a reliable reference in order to study their toxicological potency, and CTX1B obtained from Pr. Lewis is one. The assay is not intended to identify the toxins present.

We have added the information that CTX1 is not produced by microalgae and the suggested reference by the reviewer, in lines 260-266: “Obtaining CTX purified standards from microalgae is a very difficult task since the CTX production in microalgae is often very low. For this reason, the toxicological evaluation often is carried out with CTX standards purified from fish. It has been found that CTX4A, CTX4B, and CTX3C are produced by the alga, and they are oxidized to the analogsCTX1B, 52-epi-54-deoxyCTX1B, 54-deoxyCTX1B, 2-hydroxyCTX3C, and 2,3-dihydroxyCTX3C (Ikehara et al., 2017). In the present study, the reference molecule was CTX1B (Yogi et al., 2014; Lewis et al. 1991), it’s a typically CTX found in large carnivorous fish in the Pacific Ocean and which has never been found in microalgae.”

Ikehara, T., Kuniyoshi, K., Oshiro, N., Yasumoto, T., 2017. Biooxidation of ciguatoxins leads to species-specific toxin profiles. Toxins (Basel). 9, 11–14. https://doi.org/10.3390/toxins9070205

Lewis, R.J., Sellin, M., Poli, M.A., Norton, R.S., MacLeod, J.K., Sheil, M.M., 1991. Purification and characterization of ciguatoxins from moray eel (Lycodontis javanicus, Muraenidae). Toxicon 29, 1115–1127. https://doi.org/10.1016/0041-0101(91)90209-A

Yogi, K., Sakugawa, S., Oshiro, N., Ikehara, T., Sugiyama, K., Yasumoto, T., 2014. Determination of toxins involved in ciguatera fish poisoning in the Pacific by  LC/MS. J. AOAC Int. 97, 398–402. https://doi.org/10.5740/jaoacint.sgeyogi

The authors mention they monitored MTX-1 using the MRM transition m/z 482 that represents [M-2H]2->[M-2H]2-. This is incorrect. The mass of MTX-1 is 3,425.9 Da as the disodium salt. The doubly charged pseudoMRM to monitor is m/z 1689.6.

Yes, it is true. It was a mistake. The molecular mass of MTX-1 is 3,425.9 Da.

Reviewer 3 Report

The revised paper is improved but given the importance of control data in this study the authors need to present a Table of data (means +/– SEM; N=?) with and without O/V addition for each of the samples analysed for CTX, plus reference CTX, to give the reader a clearer view of toxicity determinations.

The authors need to provide more information on the selectivity of the antibodies for different CTX to ascertain which CTX are detected, and include a comment that only CTX recognised by both antibodies will be detected.

The authors should reference the papers describing the CTX used as reference.

Author Response

REVIEWER 3.

The revised paper is improved but given the importance of control data in this study the authors need to present a Table of data (means +/– SEM; N=?) with and without O/V addition for each of the samples analysed for CTX, plus reference CTX, to give the reader a clearer view of toxicity determinations.

We have added the supplementary table S3 with this information. We have mentioned the table in lines 280-282, and we have attached it.

The authors need to provide more information on the selectivity of the antibodies for different CTX to ascertain which CTX are detected and include a comment that only CTX recognised by both antibodies will be detected.

We have added more information in lines: 370 and 374. Not it reads: “The role of the antibodies in the immunosensing tool is not to confirm the presence of individual CTXs congeners as in instrumental analysis techniques, but to screen the presence of compounds with wings structurally similar to those of the four CTX targets (CTX1B, 54-deoxyCTX1B, CTX3C and 51-hydroxyCTX3C). Therefore, and because of the sandwich format of the assay, the analysis is indicating the presence of compounds structurally related to two series of CTXs congeners, although no evidence can be obtained about which specific CTXs congeners are present. It is important to mention that the strategy will detect only the CTXs recognised by the antibodies (the four major CTX congeners previously mentioned), but it is also important to highlight that the antibodies have been demonstrated to not cross-react with brevetoxin A, brevetoxin B, okadaic acid and maitotoxins.“

The authors should reference the papers describing the CTX used as reference.

The reference has been added in line 448-450.